# Editable Proof Sketch for Automated Theorem Proving

**Zi-Kai Xiao** [1]  **Han-Zheng Wang** [2]  **Meng-Hao Guo** [1]  **Shi-Min Hu** [1]  **Shing-Tung Yau** [3]

## Abstract

As large language models (LLMs) improve in mathematical reasoning and formal understanding, a promising approach for automated theorem proving (ATP) is to enable LLMs construct proof sketches, which plan a high-level proof strategy and decompose complex theorems into independently provable subgoals. However, most existing proof sketches are immutable. As a result, any revision typically requires rebuilding the entire sketch, which discards already proved subgoals and bring additional cost. In this paper, we address this limitation by introducing EditableSketch, an editable proof-sketch structure that supports in-place edits for error correction and further subgoal decomposition while preserving previously proved subgoals. Building on EditableSketch, we introduce SketchRefine, a proof-generation framework for ATP by iteratively refining proof sketches through localized, incremental edits. Experiments show that our method not only reduces the cost of the proof process, but also achieves superior performance. For example, our method realizes 76.0% pass rate on FormalMath-Lite (+14.1% vs. DeepSeek-Prover-V2-671B). Meanwhile, compared with Hilbert, our method significantly reduces token overhead while achieving comparable performance.

## 1. Introduction

With the continued development of proof assistants such as Isabelle (Paulson, 1994) and Lean 4 (de Moura & Ullrich, 2021), ATP is attracting growing attention. A widely adopted paradigm is to use LLMs to generate proof sketches (Varambally et al., 2025; Cao et al., 2025; Zhou

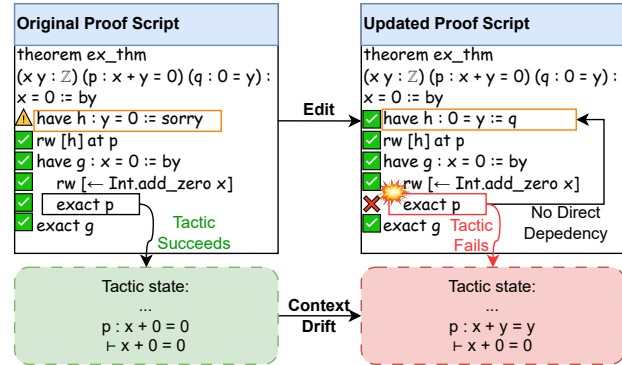

Figure 1. **Editing Fragility of Lean 4 Proof Scripts.** A local modification in a Lean 4 proof script (*e.g.,* rewriting a "have" statement) can silently alter the surrounding proof context, causing downstream tactics to fail, even if they appear to have no direct dependency on the modified line. It highlights the high coupling and poor editability of proof sketches written directly in Lean 4.

et al., 2025b) that guide the proof process by decomposing a complex theorem into simpler subgoals, which are subsequently solved by formal provers (Lin et al., 2025c; Ren et al., 2025). Proof sketches enable ATP systems to leverage both the high-level reasoning capabilities of LLMs and the verification capabilities of formal provers.

Despite their success, most existing proof sketches are immutable. Once constructed, a sketch is treated as a fixed structure. For example, some approaches (Varambally et al., 2025; Cao et al., 2025) adopt native Lean 4 scripts as their proof sketches. However, as illustrated in Figure 1, Lean 4 scripts are not well suited for direct editing. Under this structure, if a subgoal is incorrect or too difficult to prove, the system typically rebuilds the entire sketch, leading to several key limitations. First, rebuilding the entire proof sketch at each iteration incurs substantial token consumption. Second, this rebuild-based strategy discards all previously verified subgoals, forcing the system to repeatedly re-prove parts of the proof that have already been successfully established, rather than focusing solely on the unresolved components. This redundancy not only wastes LLM tokens but also degrades the overall performance of the model.

To address these limitations, we propose EditableSketch, an editable proof sketch structure supports in-place modification. Inspired by the *term-style* proof paradigm in Lean

[1]Department of Computer Science and Technology, Tsinghua University, Beijing, China [2]Zhili College, Tsinghua University, Beijing, China [3]Yau Mathematical Sciences Center, Tsinghua University, Beijing, China. Correspondence to: Shi-Min Hu <shimin@tsinghua.edu.cn>.

*Proceedings of the $43^{rd}$ International Conference on Machine Learning*, Seoul, South Korea. PMLR 306, 2026. Copyright 2026 by the author(s).

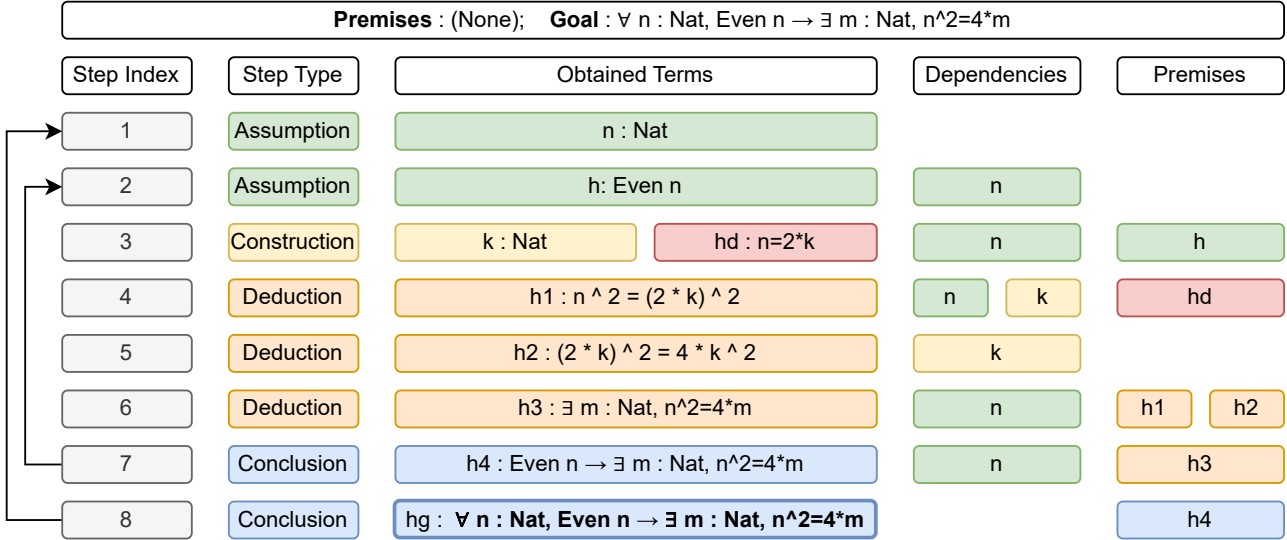

*Figure 2.* **An llustrative example of the EditableSketch structure.** Dependencies list the syntactic dependencies of nodes, while Premises are the additional prerequisites required for derivation and construction. Steps 1 and 2 are Assumption nodes that introduce assumed objects: Step 1 introduces the variable n, and Step 2 assumes Even n, which depends syntactically on the previously introduced n. Step 3 constructs a witness k together with the proposition $n = 2k$ that it must satisfy. Because the existential statement $\exists k : Nat, n = 2k$ follows from n being even, this construction is carried out under the Step 2 assumption Even n. Steps 4–6 are Deduction nodes, each constructing an intermediate propositional term. Step 7 is the Conclusion node paired with Step 2; it discharges the Step 2 hypothesis by incorporating it into the type of the term constructed at Step 6. Step 8 is the conclusion node paired with Step 1; it universally quantifies over n, incorporating the Step 1 hypothesis into the type of the term produced at Step 7.

4, EditableSketch decomposes a theorem proof into a sequence of explicit proof steps, where each step constructs new terms from available assumptions or previously derived results. By inserting, removing, or modifying individual steps, proofs can be edited in a highly localized and flexible manner. Moreover, this formulation makes dependencies between proof steps explicit: any modification only affects steps that directly depend on it, thereby avoiding the silent and non-local failures illustrated in Figure 1. Building on EditableSketch, we introduce SketchRefine, an iterative proof-generation framework for ATP. SketchRefine alternates between proving subgoals and refining the proof sketch through small, targeted edits. When a subgoal is found to be incorrect, the system directly repairs the corresponding part of the sketch. Besides, when a subgoal is correct but difficult, it is further decomposed within the existing structure. Crucially, previously proved subgoals are retained and never reproved unless affected by an edit. This design enables effective use of verifier feedback while avoiding unnecessary recomputation.

We evaluate SketchRefine on two benchmarks: MiniF2F and FormalMath-Lite. Experimental results demonstrate that our approach achieves state-of-the-art performance on both datasets. On MiniF2F-test, SketchRefine solves 99.6% of the problems, outperforming prior sketch-based methods. On FormalMath-Lite, it achieves a 76.0% pass rate, exceeding DeepSeek-Prover-V2-671B (Ren et al., 2025) by

14.1%.

In summary, our main contributions are as follows:

- We introduce EditableSketch, an editable proof sketch structure that enables localized modification while preserving previously proved subgoals.

- Buinding on EditableSketch, we propose SketchRefine, an iterative ATP framework that incrementally refines proof sketches through in-place edits guided by verifier feedback.

- Experiments demonstrate that SketchRefine achieves superior performance on both MiniF2F and FormalMath-Lite, while substantially reducing token usage compared to prior sketch-based approaches.

## 2. Related Work

### 2.1. Specialized Models for ATP

Built on interactive theorem provers such as Isabelle (Paulson, 1994) and Lean 4 (de Moura & Ullrich, 2021), numerous ATP approaches have been proposed, with large language model (LLM)-based methods showing particularly strong performance in recent work. Representative stepwise proof systems include BFS-Prover (Xin et al., 2025b), LLM-Step (Welleck & Saha, 2023), GPT-f (Polu & Sutskever, 2020), TheoremLlama (Wang et al., 2024b), InternLM2.5-StepProver (Wu et al., 2024), and Lean-STaR (Lin et al.,

2025a), while approaches that directly generate complete proofs include DeepSeek-Prover-V2 (Ren et al., 2025), the Goedel-Prover family (Lin et al., 2025b;c), and STP (Dong & Ma, 2025). These methods combine strategies such as constructing and augmenting formal training data, incorporating chain-of-thought reasoning, and leveraging reinforcement learning, achieving substantial progress on ATP tasks. Nevertheless, prior work has noted that specialized formal ATP models still exhibit a significant gap compared with state-of-the-art informal theorem-proving methods (Dekoninck et al., 2025).

## 2.2. Sketch Generation in ATP

Generating proof sketches is a common approach for enhancing formal theorem proving by leveraging the informal proving capabilities of general-purpose LLMs. Some methods (Jiang et al., 2023; Cao et al., 2025; Zhao et al., 2024) focus on improving the success rate of constructing a sketch in a single pass. However, prior work has noted that as the sampling budget increases, the standard Best-of-N sampling strategy becomes progressively less effective at test time than approaches that continuously incorporate feedback and repair the proof iteratively (Zhou et al., 2025b).

Methods represented by POETRY (Wang et al., 2024a) and Hilbert (Varambally et al., 2025) decompose the target theorem into multiple subtheorems. If the prover fails to establish some subgoals, these methods recursively further decompose the remaining subproblems until all subgoals are proved or a preset maximum recursion depth is reached. However, recursive sketch construction methods tend to lose global context at deeper recursion levels, making it difficult to identify and correct global errors.

Methods such as Delta-Prover (Zhou et al., 2025b) and LYRA (Zheng et al., 2024) typically leverage the sketch produced in the previous step together with error signals, and prompt the model to regenerate a repaired proof sketch over multiple iterations. These iterative approaches usually cannot directly edit an already constructed sketch; instead, they must regenerate the entire sketch, which leads to limited reuse of previously proved subgoals. In contrast, our method performs only localized edits to the sketch, preserving as much of the already verified content as possible.

## 3. Method

To maximally reuse successfully proved subproblems and partial proof sketches, we propose EditableSketch together with a corresponding proving method SketchRefine. This section first introduces the structure of EditableSketch, and then describes the overall procedure for proving mathematical theorems with the SketchRefine.

### 3.1. EditableSketch

Inspired by Lean 4's *term-style* proofs, we view theorem proving as a process of constructing intermediate terms from given ones until the target term is obtained. By explicitly ordering all intermediate construction steps and clearly specifying which terms are used and which terms are produced at each step, we obtain the EditableSketch proposed in this work. Section 3.1.1 and Section 3.1.2 respectively introduce the construction and editing mechanisms of EditableSketch, while Section 3.1.3 describes how an EditableSketch is compiled into Lean 4 scripts. In our implementation, we additionally design a domain-specific language (DSL) that enables a general-purpose LLM to describe the construction and modification of EditableSketch. The details of this DSL are presented in Appendix B.

### 3.1.1. BUILDING THE EDITABLESKETCH

*Term-style* proofs can be seen as a process of term construction, where each construction step may use one or more existing terms. In addition, following the idea of Delta-Prover (Zhou et al., 2025b), the EditableSketch introduces *assumption* and *conclusion* node to handle proof patterns such as case analysis and recursion. Concretely, an EditableSketch is composed of four types of nodes:

**Deduction nodes.** A deduction step introduces a single derived term together with its type. Its type may *syntactically depend* on previously available terms (*i.e.,* it can mention these terms in its statement). In addition, producing the term may require a set of *premises* that serve as logical inputs to the derivation. Deduction nodes capture forward reasoning steps that establish a new fact from existing facts.

**Construction nodes.** A construction step may introduce one or more terms simultaneously, together with the assertion that they satisfy a specified proposition (*e.g.,* constructing natural numbers $a$ and $b$ such that $a + b = 1$). Similar to deduction nodes, both the declarations of the constructed objects and the associated satisfaction condition may syntactically depend on previously introduced terms, while the existence of such objects may rely on additional premises. Construction nodes are particularly well suited for existential reasoning and for packaging structured witnesses together with their correctness conditions. Moreover, they can also be used to declare constants or to simplify and encapsulate long expressions.

**Assumption nodes.** An assumption node temporarily extends the local context by postulating the existence of a term of a given type. The assumed type may syntactically depend on previously introduced terms. Assumption nodes do not establish the assumed term; instead, they open a local scope in which subsequent reasoning may proceed under the hypothesis, enabling patterns such as contradiction,

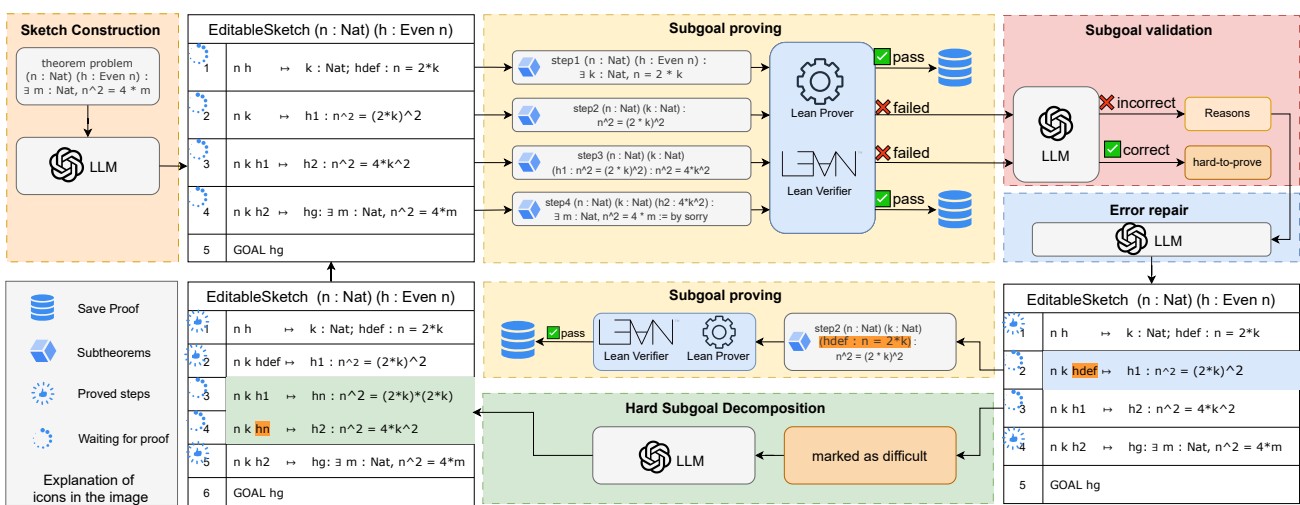

*Figure 3.* **Overall pipeline of the SketchRefine.**

conditional proofs, and recursive arguments.

**Conclusion nodes.** A conclusion node closes a scope opened by an assumption node by discharging the corresponding hypothesis. It produces a term representing a conditional statement: if the assumed term exists (or holds), then the term derived within the scope can be obtained. In this way, the conclusion nodes explicitly record the dependency of the derived result on the discharged assumption, ensuring that local reasoning does not leak outside its intended scope.

Figure 2 presents a representative instance of a EditableSketch structure. In the figure, Dependencies and Premises correspond, respectively, to the syntactic dependencies and logical premises introduced above. The key distinction is that a syntactic dependency refers to objects that are implicitly assumed to already exist when writing the item in Lean 4 (*i.e.,* they must be defined earlier in the EditableSketch for the term to type-check), whereas a logical premise is used only inside the proof that constructs the term. For example, the statement a = c syntactically depends on a and c, while a = b and b = c are the premises required to derive it. Dependencies and Premises are handled differently when compiling EditableSketch into subproblems and Lean 4 sketches; we detail this in Section 3.1.3.

Moreover, assumption and conclusion nodes must occur in matched pairs. Such pairs may be nested but must not cross. Each matched assumption-conclusion pair defines a scope; nodes inside the scope are invisible to nodes outside. For instance, in Figure 2, Step 2 with Step 7 and Step 1 with Step 8 each form a scope. From the perspective of Step 8, the nodes from Step 2 through Step 6 are not visible, while the conclusion established at Step 7 may be used by Step 8.

### 3.1.2. EDITING THE EDITABLESKETCH

Single-step edits in EditableSketch comprise insertion, update, and deletion, each operating on exactly one node. Specifically, we consider three primary edit types:

**Insertion.** Insert a new node immediately after the node with a specified index.

**Update.** Identify a node by its index and replace its entire content, including its syntactic dependencies and logical premises.

**Deletion.** Identify a node by its index and remove it from the EditableSketch structure.

A complete edit may consist of multiple single-step edits that are executed sequentially. After all single-step edits in a complete edit have been applied, we perform a syntactic consistency check, which is detailed in Section 3.1.3.

### 3.1.3. COMPILING THE EDITABLESKETCH

We refer to the process of converting a EditableSketch into its corresponding Lean 4 script as EditableSketch compilation. Concretely, a EditableSketch is compiled into two parts: (i) a collection of subtheorems to be proved, and (ii) a Lean 4 fragment that proves the original theorem by invoking these subtheorems.

As a prerequisite, we run a structural correctness check on the EditableSketch to ensure that (i) every node mentions only syntactic dependencies and logical premises that have been introduced earlier and are in scope; (ii) Assumption and Conclusion nodes form properly matched pairs without crossings; and (iii) the final constructed term has the target type required by the problem.

Given a structurally correct EditableSketch, we generate

subtheorems from the graph. Each Deduction node and each Construction node gives rise to one subtheorem, whose premises are the union of its syntactic dependencies, logical premises, and indirect syntactic dependencies. The notions of syntactic dependency and logical premise are introduced in Section 3.1.1; we define indirect syntactic dependencies as the union of the syntactic dependencies of all logical premises. Since natural-language models often struggle to identify indirect dependencies reliably, we require the model to annotate only direct syntactic dependencies and logical premises when generating the EditableSketch; the compiler then recovers indirect dependencies by aggregating the dependencies of the premises.

In parallel, we synthesize the Lean 4 proof sketch for the original goal, primarily via term construction. A Deduction node is translated into a "`have`" statement that constructs the corresponding term; a Construction node is translated into an "`rcases`" statement that extracts a tuple of terms from an existential proposition; and each matched Assumption–Conclusion pair is translated into a "`have ... := by`" block that opens a temporary context, with "`intro`" introducing the assumed term into that context.

Finally, during EditableSketch compilation we invoke Lean 4 for syntax and type checking to ensure that the generated subtheorems (their proofs are temporarily replaced by "`sorry`") together with the Lean 4 fragment, are accepted by the Lean 4 compiler. Once all subtheorems are proved, concatenating their proofs with the Lean 4 fragment yields a complete proof of the original theorem.

### 3.2. SketchRefine

Given a formalized theorem, SketchRefine attempts to construct a proof using EditableSketch. It first invokes a specialized prover to generate eight direct proof candidates; if any is verified, it is accepted as the final proof. Otherwise, a general-purpose reasoning LLM is prompted to produce an initial EditableSketch, which is then iteratively refined by alternating between subgoal proving and feedback-driven revision. Section 3.2.1 details the core pipeline of SketchRefine (Figure 3). To improve robustness against outdated Lean 4 knowledge in general-purpose LLMs and formalization errors in the dataset, we further introduce the *Theorem Retrieval* and *Refutation* stages (Sections 3.2.2 and 3.2.3). The complete pipeline is described in Appendix C.

#### 3.2.1. OVERALL PIPELINE

**Step 0 (Sketch Construction).** Given the target theorem, we prompt a general-purpose LLM to synthesize an EditableSketch that satisfies the constraints in Section 3.1.1. Each candidate EditableSketch is processed by a lightweight compiler that parses the DSL and reports precise diagnostic messages when the output violates the specification. Any

syntax, well-formedness, or type-checking errors are fed back to the LLM for revision. This stage terminates once we obtain a well-formed EditableSketch whose induced subgoals are all accepted by the Lean 4 verifier under "`sorry`" placeholders.

**Step 1 (Subgoal proving).** For each subgoal generated by the EditableSketch, we ask the prover to produce eight candidate proofs and retain any proof that is accepted by the verifier as a valid solution for the corresponding node. If none of the eight candidates passes verification, the corresponding subgoal is marked as *hard-to-prove*.

**Step 2 (Subgoal validation).** For the EditableSketch produced after Step 1, we collect all subgoals marked as *hard-to-prove* and submit them to the general-purpose LLM for plausibility checking. The LLM is asked to assess whether each subgoal is correct and, if not, to provide a detailed explanation of the error. If at least one subgoal is identified as *incorrect*, we proceed to Step 3 to repair the EditableSketch; otherwise, we proceed to Step 4 to further decompose the hard subgoals.

**Step 3 (Error repair).** When one or more subgoals are judged to be incorrect, we aggregate the corresponding error reports and instruct the LLM to propose edits in the format specified in Section 3.1.2. We then validate the proposed edits by (i) checking syntactic well-formedness, (ii) ensuring that the revised EditableSketch satisfies the structural constraints, and (iii) re-running Lean 4 verification on all induced subtheorems with "`sorry`" placeholders. Any resulting diagnostics are returned to the model to trigger another revision. After a successful repair, we invalidate cached proofs for all nodes whose associated subgoals have changed and return to Step 1. Importantly, only nodes without an already-verified proof are re-attempted; previously verified nodes are not reproved.

**Step 4 (Hard Subgoal Decomposition).** If all unsolved subgoals are judged correct in Step 2, we treat them as legitimate but challenging. We then identify subgoals that still lack a verified proof and instruct the general-purpose LLM, again following the format in Section 3.1.2, to refine the EditableSketch so as to decompose these subgoals into simpler ones. The resulting edits are processed using the same validation pipeline as in Step 3. We then return to Step 1 and continue attempting proofs only for nodes that still have no recorded valid proof.

#### 3.2.2. THEOREM RETRIEVAL

Since Lean 4 and its accompanying Mathlib library are under active development, the knowledge of a general-purpose LLM may lag behind the current ecosystem. This mismatch can induce hallucinations, such as invoking outdated lemmas or referencing non-existent theorems. To mitigate

these issues, we incorporate a theorem-retrieval stage that searches for lemmas relevant to each subgoal and supplies them to the general-purpose LLM, thereby improving the efficiency and reliability of subgoal decomposition.

The theorem-retrieval stage is placed immediately before Step 4 (Hard Subgoal Decomposition). When theorem retrieval is enabled, we decompose only one hard subgoal at a time. Specifically, we first ask the general-purpose LLM to propose at most five keyword queries. We then apply a theorem-retrieval model to return three candidate theorems per query. Given these candidates, the general-purpose LLM selects up to five theorems that it deems most relevant. After retrieval, we additionally prompt the LLM to attempt a direct proof of the hard subgoal, aiming to capture cases where the subgoal is essentially a straightforward application of an existing Mathlib lemma but remains difficult for the prover to solve. If a verified proof is produced, we proceed to the next hard subgoal; otherwise, we continue with Step 4 and instruct the model to decompose the subgoal while leveraging the retrieved theorems.

Furthermore, any newly generated subgoals inherit the set of theorems associated with the parent hard subgoal. If a descendant subgoal requires further decomposition, the candidate theorem set used in retrieval consists of both (i) the at most fifteen candidates returned by the retrieval model and (ii) the inherited theorems. This design prevents high-utility theorems from being lost across successive decompositions.

### 3.2.3. REFUTATION-GUIDED HANDLING OF UNPROVABLE GOALS WITH PIPELINE RESTARTS.

Due to formalization mistakes and related issues, a subset of problems in the test set may be unprovable. In such cases, the system can repeatedly enter Step 3 (Error repair) without making progress. To address this failure mode, in Step 3 we additionally ask the model to assess whether the original theorem is unprovable. When the model determines that the theorem is indeed unprovable, it is required to provide a refutation: a derivation showing that the original theorem leads to a contradiction.

Once unprovability is detected in Step 3, we restart the pipeline and switch from proof to refutation. Concretely, we replace the target theorem with a goal of the form "assume the original theorem holds; prove `False`," and provide the model with the contradiction witness identified in Step 3. If the refutation goal is again judged unprovable, which may indicate that the previous contradiction was spurious or that the original theorem is actually provable, we restart the pipeline once more and revert to proving the original theorem.

Our refutation step provides a **sound** certificate of the problem's unprovability, with correctness guaranteed by the con-

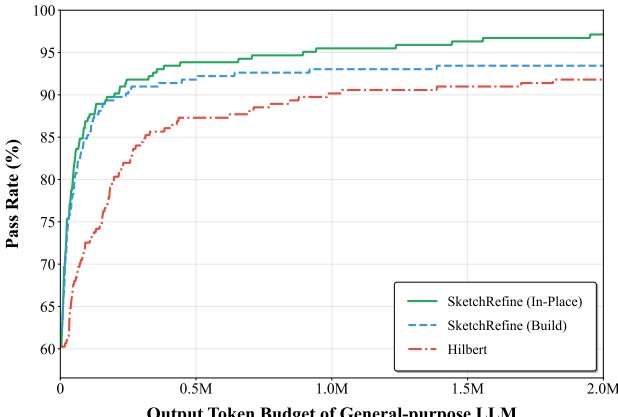

*Figure 4.* **Pass rates of Hilbert and SketchRefine vs. the general-purpose LLM output token budget.** Both methods use DeepSeek-V3.2 as the general-purpose LLM and DeepSeek-Prover-V2-7B (non-CoT) as the prover, and all results are evaluated on MiniF2F-test. Hilbert results are reproduced using the official implementation. SketchRefine (Build) synthesizes a complete EditableSketch for each decomposed subtheorem and then integrates it into the original EditableSketch, whereas SketchRefine (In-Place) edits the original EditableSketch directly during decomposition.

sistency of Lean 4.

we compare how the number of additional premises affects the specialized prover's pass rate. Specifically, we collect all sub-theorems produced by the final EditableSketch on MiniF2F-test and add varying numbers of random irrelevant premises to these sub-theorems. Based on the experimental results, we find that as the number of added irrelevant premises increases, the specialized prover's pass rate drops significantly. With a larger sampling budget, the gap narrows slightly, but additional irrelevant premises still noticeably reduce the prover's pass rate. Therefore, ensuring that subproblems contain only the necessary premises clearly helps improve the pass rate.

## 4. Experiments

### 4.1. Experimental Settings

All experiments are conducted with Lean 4 (v4.19.0) and Mathlib (v4.19.0). We use Kimina Lean Server (Santos et al., 2025) as the proof-checking backend for verifying Lean 4 proofs. For subgoal proving, we adopt DeepSeek-Prover-V2-7B (Ren et al., 2025); for general-purpose LLM reasoning, we use Gemini 2.5 Pro and DeepSeek-V3.2; for theorem retrieval, we adopt LeanExplore (Asher, 2025). For evaluation, we count a problem as solved (and include it in the pass rate) if either the original theorem is formally proved or a certified refutation is produced by our refutation strategy (*i.e.,* deriving "`False`" from assuming the statement), so unprovable instances resolved via refutation

*Table 1.* **Performance on MiniF2F-test.** Note: The result of Gemini 2.5 pro is sourced from Zhou et al. (2025b)

| Method | Settings | Pass Rate |
| --- | --- | --- |
| Gemini 2.5 Pro | pass@16384 | 49.1% |
| Leanabell-Prover (Zhang et al., 2025) | pass@128 | 61.1% |
| STP (Dong & Ma, 2025) | pass@25600 | 67.6% |
| DeepSeek-Prover-V2-7B (Ren et al., 2025) | pass@8192, non-CoT | 75.0% |
| | pass@8192, CoT | 82.0% |
| DeepSeek-Prover-V2-671B (Ren et al., 2025) | pass@8192, non-CoT | 78.3% |
| | pass@8192, CoT | 88.9% |
| Kimina-Prover-72B (Wang et al., 2025) | pass@1024 | 87.7% |
| | w/ TTRL | 92.2% |
| Goedel-Prover-V2-8B (Lin et al., 2025c) | pass@8192 | 90.2% |
| | pass@1024, w/ self-correction | 89.3% |
| Goedel-Prover-V2-32B (Lin et al., 2025c) | pass@8192 | 92.2% |
| | pass@1024, w/ self-correction | 92.6% |
| BFS-Prover-V2-32B (Xin et al., 2025c) | w/o Planner | 86.1% |
| | w/ Planner | 95.1% |
| DSP+ (Cao et al., 2025) | DeepSeek-R1 + DeepSeek-V3-0324 + BFS-Prover-7B | 80.7% |
| | ensemble | 83.6% |
| Delta-Prover (Zhou et al., 2025b) | Gemini 2.5 pro | 95.9% |
| Hilbert (Varambally et al., 2025) | Gemini 2.5 pro + DeepSeek Prover-V2-7B non-CoT | 98.4% |
| | Gemini 2.5 pro + Goedel-Prover-V2-32B | 99.2% |
| SketchRefine (w/o refutation) | DeepSeek-V3.2 + DeepSeek-Prover-V2-7B non-CoT | 96.7% |
| SketchRefine (w/ refutation) | DeepSeek-V3.2 + DeepSeek-Prover-V2-7B non-CoT | 97.5% |
| SketchRefine (w/o refutation) | Gemini + DeepSeek-Prover-V2-7B non-CoT | 98.8% |
| SketchRefine (w/ refutation) | Gemini + DeepSeek-Prover-V2-7B non-CoT | **99.6%** |

contribute to the pass rate.

**MiniF2F** (Zheng et al., 2022) is a benchmark of competition-level high-school mathematics problems, where both the validation and test splits contain 244 fully formalized theorems. The statements are drawn from sources such as AIME and IMO, as well as classical textbook results, and many instances are highly challenging. We evaluate our method and conduct ablations on the test split MiniF2F-test. On MiniF2F-test, we impose a budget of 10M output tokens from the general-purpose LLM per problem. The Lean 4 MiniF2F-test version we used comes from the open-source repository of Lin et al. (2025c).

**FormalMath** (Yu et al., 2025) covers a broad range of mathematical domains and difficulty levels, from high-school Olympiad problems to undergraduate-level theorems, including algebra, applied mathematics, calculus, number theory, and discrete mathematics. We evaluate on FormalMath-Lite, a subset containing 425 problems. On FormalMath-Lite, we cap the total number of output tokens generated by the DeepSeek-V3.2 at 2M per problem.

### 4.2. Overall Performance

We report the pass rate of SketchRefine on MiniF2F-test and FormalMath-Lite in Table 1 and Table 2, respectively. The results show that SketchRefine achieves state-of-the-art performance on both datasets.

On MiniF2F-test, SketchRefine with Gemini 2.5 Pro fails on only `imosl_2007_algebra_p6` and `imo_2001_p6`, achieving a pass rate of 99.2%, which exceeds the Hilbert (Varambally et al., 2025) method with the same configuration by 0.8%. After switching to Gemini 3.0 Pro Preview, SketchRefine successfully solves `imo_2001_p6`, thereby reaching state-of-the-art performance on this dataset. When using the open-source model DeepSeek-V3.2, SketchRefine attains a pass rate of 97.5% on MiniF2F-test, outperforming Delta-Prover (Zhou et al., 2025b) with the closed-source Gemini 2.5 Pro.

On FormalMath-Lite, SketchRefine with the open-source DeepSeek-V3.2 surpasses the previous best method, DeepSeek-Prover-671B, by 14.1% in pass rate. We attribute the superior pass rate of SketchRefine to the editability of EditableSketch, its effective use of verifier feedback, and its ability to avoid discarding subproblems whose proofs have already been completed. We provide a more detailed

*Table 2.* **Performance on FormalMath-Lite.** The experimental results for DS-Prover-v1.5-RL (Xin et al., 2025a), InternLM-V2.5 (Wu et al., 2024), BFS-Prover (Wu et al., 2024), Kimina-Prover-7B (Wang et al., 2025), STP (Dong & Ma, 2025), Goedel-Prover (Lin et al., 2025b) and DS-ProverV2 (Ren et al., 2025) are sourced from Yu et al. (2025), while the experimental results for Goedel-Prover-V2-8B (Lin et al., 2025c) and Spark-Prover-X1-7B are sourced from Zhou et al. (2025a). Note: DS is short for DeepSeek. SketchRefine uses DeepSeek-V3.2 as the general-purpose LLM and DeepSeek-Prover-V2 7B as the prover model.

| Method | Pass Rate |
|---|---|
| DS-Prover-V1.5-RL (32×32×100) | 17.4% |
| InternLM-V2.5 (32×32×100) | 25.7% |
| BFS-Prover (32×32×100) | 45.9% |
| Kimina-Prover-7B (pass@32) | 48.9% |
| STP (pass@3200) | 53.2% |
| Goedel-Prover-V2-8B (pass@32) | 55.3% |
| Spark-Prover-X1-7B (pass@32) | 59.8% |
| Goedel-Prover (pass@3200) | 49.4% |
| DS-ProverV2-7B (CoT, pass@3200) | 55.1% |
| DS-ProverV2-671B (CoT, pass@3200) | 61.9% |
| SketchRefine (w/o refutation) | 69.2% |
| SketchRefine (w/ refutation) | **76.0%** |

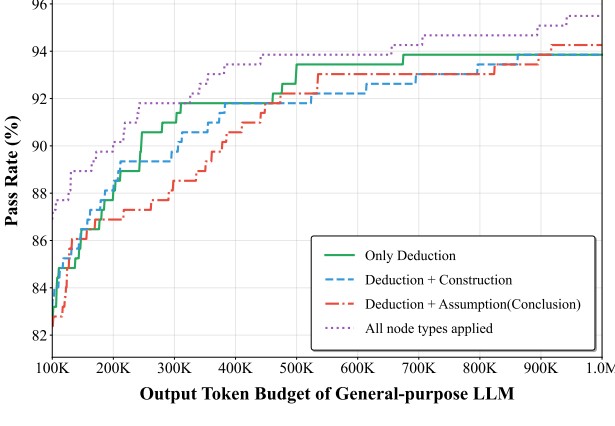

*Figure 5.* **Effect of Node Types.** We evaluate the pass rate when only a subset of node types is enabled by modifying the prompting strategy. In particular, Assumption and Conclusion nodes are required to appear in pairs; we denote the support for such paired nodes as Assumption(Conclusion). All experiments are conducted using DeepSeek-V3.2 as the general-purpose LLM and DeepSeek-Prover-V2-7B (non-CoT) as the prover.

description of the hyperparameters of SketchRefine, along with the LLM overhead, in Appendix D.

In the MiniF2F-test dataset, we identify two problems (`amc12a_2021_p25` and `imo_1982_p1`) that are unprovable under the given formalization, and we provide a brief analysis of these cases in Appendix A. The dataset used by the Hilbert (Varambally et al., 2025) differs slightly from ours; after adjusting the formalizations of these two problems, Hilbert is able to complete their proofs.

### 4.3. Ablation Study

**Ablation study on subgoal handling.** We consider two strategies for subgoal decomposition: (i) treating each subgoal as a new standalone problem and constructing a corresponding proof sketch (Build strategy), and (ii) treating subgoal decomposition as an in-place modification of the original proof sketch (In-Place strategy). Figure 4 compares the pass rates of these two strategies. The results show that the In-Place strategy consistently and substantially outperforms the Build strategy, and the gap widens as problem difficulty increases (as reflected by higher token usage). We attribute this advantage to the fact that the In-Place strategy preserves the global proof plan during decomposition; moreover, when decomposing similar subgoals, the LLM can reuse and cross-reference information across subgoals within the same evolving sketch. In contrast, the Build strategy discards global context, increasing the cost of producing effective decompositions. Unless oth-

erwise specified, we use the In-Place strategy as the default subgoal-decomposition method for all other experiments.

**Comparison with Hilbert.** In Figure 4, we also compare the pass rates of SketchRefine and Hilbert (Varambally et al., 2025) under matched token budgets. Regardless of the subgoal-decomposition strategy used by SketchRefine, SketchRefine consistently achieves higher pass rates than Hilbert at the same token cost, indicating substantially better token efficiency on the general-purpose LLM. We attribute this advantage to SketchRefine's incremental refinement strategy, which preserves intermediate progress and avoids re-solving subproblems that have already been addressed. In contrast, Hilbert performs recursive subgoal decomposition and, when the proof sketch encounters an error, must reconstruct the entire sketch, resulting in higher token overhead on the general-purpose LLM.

**Ablation Study on Node Types.** In Figure 5, we examine the impact of node types on the pass rate. The results show that removing either Construction nodes or Assumption(Conclusion) nodes consistently decreases the pass rate. The setting that retains only Deduction nodes achieves a pass rate comparable to that of removing only Construction nodes or only Assumption(Conclusion) nodes. We interpret these findings as evidence that defining four node types enables more flexible construction and editing of proof sketches, thereby achieving a higher pass rate under the same token-budget constraint.

# 5. Conclusion

We introduce EditableSketch and its associated automated theorem-proving system, SketchRefine. By editing proof sketches in place, SketchRefine corrects errors and decomposes subtheorems without rebuilding the sketch or discarding verified subgoals. On the MiniF2F-test and FormalMath-Lite benchmarks, SketchRefine achieves state-of-the-art results, with a 99.6% pass rate on MiniF2F and 76.0% on FormalMath-Lite, surpassing the previous best by 14.1% on the latter. Despite these improvements, EditableSketch has limitations: in principle, syntactic dependencies could be extracted directly by the Lean 4 compiler rather than inferred by an LLM. Future work will explore compiler-based tooling to automatically recover such dependencies, allowing the LLM to focus on mathematical reasoning.

# Acknowledgements

This work was supported by Fundamental and Interdisciplinary Disciplines Breakthrough Plan of the Ministry of Education of China (No. JYB2025XDXM101), the National Natural Science Foundation of China (project No. 62495061, 623B2057), the Research Grant of Tsinghua-Tencent Joint Laboratory for Internet Innovation Technology.

# Impact Statement

This paper presents work whose goal is to advance the field of Machine Learning. There are many potential societal consequences of our work, none which we feel must be specifically highlighted here.

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

# A. Analysis of Unprovable Problems in MiniF2F-test

In our experiments, we found that `amc12a_2021_p25` and `imo_1982_p1` are unprovable. The concrete statements of these two problems are as follows:

```
theorem imo_1982_p1 (f : ℕ → ℕ)
    (h₀ : ∀ m n, 0 < m ∧ 0 < n → f (m + n) − f m − f n = 0 ∨ f (m + n) − f m − f n = 1)
    (h₁ : f 2 = 0) (h₂ : 0 < f 3) (h₃ : f 9999 = 3333) : f 1982 = 660 := by
theorem amc12a_2021_p25 (N : ℕ) (f : ℕ → ℝ)
    (h₀ : ∀ n, 0 < n → f n = (Nat.divisors n).card / n ^ ((1 : ℝ) / 3))
    (h₁ : ∀ (n) (_ : n ≠ N), 0 < n → f n < f N) : (List.sum (Nat.digits 10 N)) = 9 := by
```

For `imo_1982_p1`, the key issue arises from the fact that subtraction on natural numbers in Lean 4 is truncated (*i.e.,* $a - b = 0$ whenever $a \leq b$). We construct a function $f$ such that $f(n) = 661$ when $n = 1982$, and $f(n) = \lfloor n/3 \rfloor$ otherwise. This function $f$ satisfies all the assumptions of the problem, yet $f(1982) \neq 660$, which directly leads to a contradiction.

For `amc12a_2021_p25`, the problem statement imposes no restriction on $N$, nor does it specify any constraint on the value of $f(n)$ at $n = 0$. As a result, $f(0)$ can be chosen to be sufficiently large so that $f(0)$ attains the maximum value of the function. Since $\frac{d(n)}{n^{\frac{1}{3}}}$ is finite for all positive integers $n$ (where $d(n)$ is the number of factors), we may further define $f(0)$ to be strictly greater than this maximum. Under this construction, the function $f$ together with $N = 0$ satisfies the assumptions $h_0$ and $h_1$, while the sum of digits of 0 is 0, which contradicts the conclusion of the problem.

We provide the complete Refutation Lean 4 scripts for both problems in the supplementary material.

# B. The DSL of EditableSketch

We design a domain-specific language (DSL) for EditableSketch to describe both its construction and subsequent modifications.

## B.1. DSL for Constructing EditableSketch

Based on Section 3.1.1, each node in EditableSketch corresponds to a single line in the DSL.

**Deduction nodes.** These are written as `STEP [index] DEDUCTION [name] : [content] REFER [names...] FROM [names...]`, indicating the derivation of an item named "`name`" with type "`content`".

**Construction nodes.** These are written as `STEP [index] CONSTRUCTION ([name1] : [Type1]), ([name2] : [Type2]) ... SATISFY [prop_name] : [content] REFER [names...] FROM [names...]`, indicating the construction of a collection of items that satisfy the condition named "`prop_name`" with specification "`content`".

**Assumption nodes.** These are written as `STEP [index] ASSUMPTION [name] : [type] REFER [names...]`, indicating the assumption of the existence of an item named "`name`" with type "`type`".

**Conclusion nodes.** These are written as `STEP [index] CONCLUSION [name] FROM ASSUMPTION [name_a] DEDUCTION [name_d]`, indicating the construction of a conclusion item named "`name`", expressing that "`name_d`" can be derived whenever "`name_a`" exists.

**Goal.** This is written as `STEP [index] GOAL [name]`, declaring that the item named "`name`" is the final conclusion to be constructed for the theorem.

In all DSL statements, the identifiers following "`REFER`" denote syntactic dependencies, while those following "`FROM`" denote logical premises; all identifiers are space-separated.

## B.2. DSL for Editing EditableSketch

Modifications to EditableSketch can be viewed as edits applied to its corresponding construction DSL. Building upon Section 3.1.2, we further introduce an additional modification type, "`ADJUST`", which is used to represent changes that only affect logical dependencies.

**Insertion.** `NEW AFTER [pre_index] [sentence]` inserts a new statement "`sentence`" after the statement indexed by "`pre_index`".

**Update.** `UPDATE [sentence]` locates the statement with the corresponding index and replaces it with "`sentence`".

**Deletion.** `DELETE [index]` removes the statement indexed by "`index`".

**Adjust.** `ADJUST STEP [index] FROM [names...]` replaces the "`FROM`" clause of the statement indexed by "`index`" with the specified list "`[names...]`".

## C. Detailed Pipeline of SketchRefine

During execution, the SketchRefine pipeline maintains a mutable EditableSketch, and treats every modification to EditableSketch as an independent step. Concretely, in addition to the stages defined in Section 3.2.1, we introduce the following steps:

**Pre-proving.** Before performing any sketch construction, SketchRefine first invokes a specialized prover up to eight times to attempt to directly prove the target sub-theorem. If the proof succeeds, no further processing is required.

**Initialization.** For theorems that cannot be discharged during the pre-proving stage, we first parse the premises and the conclusion of the problem, and then invoke a general-purpose LLM to analyze the syntactic dependency structure among these premises.

**Theorem Retrieval.** Given the earliest subproblem in EditableSketch whose proof fails, the theorem retrieval stage prompts the LLM to produce at most five keywords, corresponding to the names of relevant theorems or definitions in the MATHLIB library, or to commonly used names of the target theorem. We then apply the LeanExplore to identify the top three most relevant theorems or definitions associated with these keywords, which are subsequently added to the set of related theorems for this subproblem.

**Theorem Selection.** For the same earliest failing subproblem in EditableSketch, the theorem selection stage prompts the LLM to choose at most five theorems that are most relevant for proving this subproblem from its associated set of related theorems, discarding the remaining ones.

**General-Proposed LLM Proving.** For the earliest failing subproblem in EditableSketch, the general-proposed LLM proving stage prompts the LLM to directly generate Lean 4 proof code for this subproblem, conditioned on the selected related theorems.

Based on these steps, we present the pseudocode of the SketchRefine pipeline in Algorithm 1.

## D. Additional Experimental Details

### D.1. Hyperparameter Configuration of SketchRefine

Invoking the general-purpose LLM only once does not necessarily yield a DSL that both conforms to the specification in Section B and is free of Lean 4 errors. When the generated DSL contains errors, we provide the corresponding error messages to the general-purpose LLM and prompt it to produce a corrected version. For the **Sketch Construction** stage, we allow up to four iterative correction attempts; for the **Error Repair** and **Hard Subgoal Decomposition** stages, we allow up to three iterative correction attempts. If the maximum number of correction iterations is exceeded, SketchRefine restarts these stages.

The **Theorem Retrieval**, **Theorem Selection**, and **General-Model Proving** stages introduced in Section C each permit at most three iterative correction attempts by the LLM. If the maximum number of correction iterations is exceeded, SketchRefine skips the corresponding stage.

### D.2. LLM Overhead

Figure 6 illustrates the relationship between the pass rate of SketchRefine and the general-purpose LLM cost on MiniF2F-test and FormalMath-Lite, while the average cost is reported in Table 3. For comparison, Hilbert+Gemini 2.5 Pro+Goedel-Prover-V2-32B gets a 99.2% pass rate on MiniF2F-test, with an average Output+Prompt cost of 2.3M tokens per problem requiring decomposition.

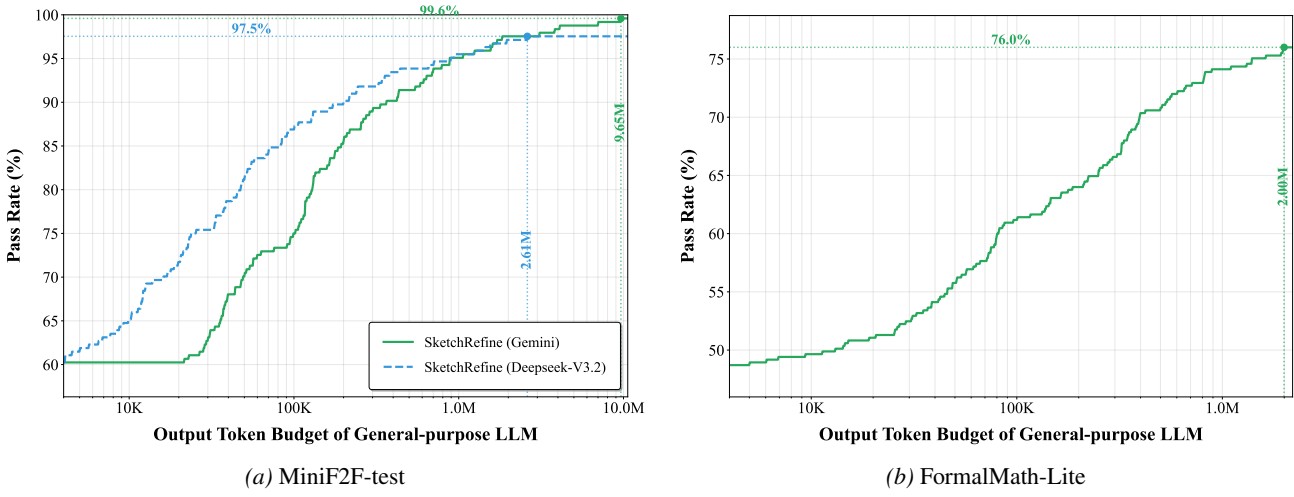

*(a)* MiniF2F-test              *(b)* FormalMath-Lite

*Figure 6.* **Pass Rate (vs) Output Budget of General-Proposed LLM.**

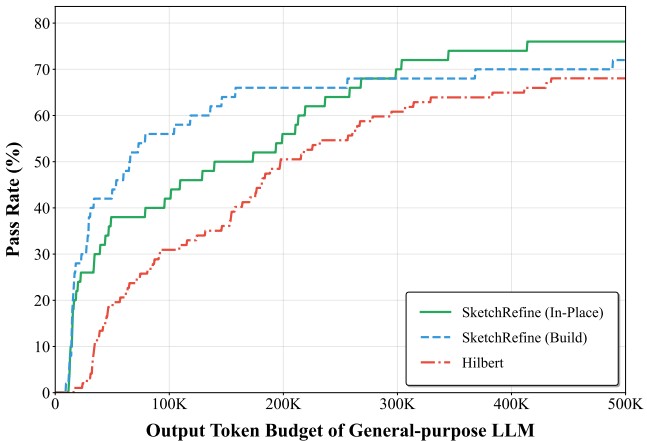

*Figure 7.* **Ablation Study on Theorem Retrieval.** Experiments are conducted on 50 randomly sampled MiniF2F-test instances that cannot be proved by Pre-Prover. *Parallel Decomposition* and *Sequential Decomposition* denote two strategies in the Hard Subgoal Decomposition stage: in the former, a general-purpose LLM decomposes all hard subgoals simultaneously, whereas in the latter, it decomposes only the first hard subgoal.

| LLM | Dataset | Pass Rate | Avg. Output | Avg. Output+Prompt | Max. Output+Prompt |
|---|---|---|---|---|---|
| Gemini | MiniF2F-test | 99.6 | 547K | 778K | 15.3M |
| DeepSeek-V3.2 | MiniF2F-test | 97.5 | 197K | 291K | 3.95M |
| DeepSeek-V3.2 | FormalMath-Lite | 76.0 | 306K | 442K | 2.91M |

*Table 3.* **Token Cost Statistics of General-Proposed LLM of SketchRefine.** The average value is calculated over problems that cannot be proven by the Pre-proving Stage.

### D.3. Ablation Study on Theorem Retrieval

In Figure 7, we evaluate the impact of the theorem-retrieval module and plot the proof pass rate on MiniF2F-test as a function of the number of output tokens produced by the general-purpose LLM. Due to the inheritance mechanism in theorem retrieval, the Hard Subgoal Decomposition step with theorem retrieval can decompose only one subgoal per iteration. The results indicate that decomposing a single subgoal per iteration yields pass rate comparable to decomposing all subgoals at once. The variant with theorem retrieval underperforms the one without retrieval on easier problems; however, as the problems become more challenging and the token budget increases, the retrieval-based approach matches the pass rate of the non-retrieval variant and shows a tendency to surpass it. We attribute this to the fact that for easy instances, SketchRefine can typically construct proofs without retrieval, making the benefit of theorem retrieval less pronounced. As the difficulty increases, the advantage of theorem retrieval becomes more evident, in particular because it helps SketchRefine align with the verifier's Lean 4 and Mathlib4 versions. Therefore, we retain the theorem-retrieval module in all other experiments.

---

**Algorithm 1** SketchRefine Inference Procedure

---

1: **Input:** Lean 4 goal $P$; Lean 4 header $H$; specialized prover $\mathcal{P}$; general-purpose LLM reasoner $\mathcal{R}$
2: **Output:** a verified Lean 4 proof script
3: $(ok, \pi) \leftarrow PreProve(\mathcal{P}, P, H)$
4: **if** $ok$ **then**
5:     **return** $\pi$
6: **end if**
7: $\mathcal{C} \leftarrow Initialize(\mathcal{R}, P, H)$ {problem context}
8: $state \leftarrow$ BUILD
9: $useRefutation \leftarrow$ FALSE
10: **while not** TokenBudgetExceeded($\mathcal{R}$) **do**
11:     **if** $state =$ BUILD **then**
12:         $S \leftarrow SketchConstruction(\mathcal{R}, \mathcal{C}, H)$
13:         $state \leftarrow$ PROVE
14:     **else if** $state =$ PROVE **then**
15:         $S \leftarrow SubgoalProving(\mathcal{P}, S, H)$
16:         $state \leftarrow$ VALIDATE
17:     **else if** $state =$ VALIDATE **then**
18:         $(unsolv, reason, hasErr, S) \leftarrow SubgoalValidation(\mathcal{R}, S, H)$
19:         **if** $unsolv$ **then**
20:             $\mathcal{C} \leftarrow GenerateRefutationContext(\mathcal{R}, P, reason, useRefutation)$
21:             $useRefutation \leftarrow \neg useRefutation$
22:             $state \leftarrow$ BUILD
23:         **else if** $hasErr$ **then**
24:             $state \leftarrow$ REPAIR
25:         **else**
26:             $state \leftarrow$ RETRIEVE
27:         **end if**
28:     **else if** $state =$ REPAIR **then**
29:         $S \leftarrow ErrorRepair(\mathcal{R}, S, H)$
30:         $state \leftarrow$ PROVE
31:     **else if** $state =$ RETRIEVE **then**
32:         $S \leftarrow TheoremRetrieval(\mathcal{R}, S, H)$
33:         $state \leftarrow$ SELECT
34:     **else if** $state =$ SELECT **then**
35:         $S \leftarrow TheoremSelection(\mathcal{R}, S, H)$
36:         $state \leftarrow$ SOLVE
37:     **else if** $state =$ SOLVE **then**
38:         $(solved, S) \leftarrow ReasonerProving(\mathcal{R}, S, H)$
39:         **if** $solved$ **then**
40:             $state \leftarrow$ RETRIEVE
41:         **else**
42:             $state \leftarrow$ DECOMPOSE
43:         **end if**
44:     **else if** $state =$ DECOMPOSE **then**
45:         $S \leftarrow SubgoalDecomposition(\mathcal{R}, S, H)$
46:         $state \leftarrow$ PROVE
47:     **end if**
48:     **if** AllSubgoalsProved($S$) **then**
49:         **return** CheckAndBuildFinalLeanScript($S, H$)
50:     **end if**
51: **end while**
52: **return** Failure("token budget exceeded")

---

