# OpenReview forum: "Editable Proof Sketch for Automated Theorem Proving"
_ICML.cc/2026/Conference — ICML 2026 regular_

### Official Review · Reviewer_HA8x · 2026-03-06

**Soundness:** 3
**Presentation:** 3
**Significance:** 3
**Originality:** 3
**Overall Recommendation:** 5
**Confidence:** 3

**Summary:**

This paper introduces a novel framework for automated theorem proving that leverages LLMs to construct and refine formal proofs. The authors propose EditableSketch, a new data structure for representing proof sketches that, unlike previous methods, allows for localized, in-place modifications. This structure supports error correction and subgoal decomposition without requiring a full rebuild of the proof, thus preserving previously verified components. Building on this, the paper presents SketchRefine, an iterative refinement framework that employs an LLM to generate and edit these proof sketches. The framework operates in a multi-stage pipeline that includes sketch construction, subgoal proving with a specialized prover, validation, error repair, and decomposition of difficult subgoals. The authors evaluate their method on the MiniF2F-test and FormalMath-Lite benchmarks, demonstrating state-of-the-art results. Notably, SketchRefine achieves a 99.6% pass rate on MiniF2F-test and a 76.0% pass rate on FormalMath-Lite, significantly outperforming prior sketch-based and direct-generation methods while also being more token-efficient.

**Compliance With Llm Reviewing Policy:**

Affirmed.

**Final Justification:**

I have no further concerns.

**Key Questions For Authors:**

- Could you please clarify which specific version of the Gemini model was used to generate the main results for SketchRefine that are presented in Table 1?

**Limitations:**

yes

**Strengths And Weaknesses:**

### Strengths
- The paper introduces a novel and intuitive representation for proof sketches that enables localized editing, which is a significant advantage over methods that require rebuilding the entire proof structure after a change.
- The proposed iterative refinement framework effectively combines the high-level reasoning capabilities of large language models with the rigor of specialized automated provers.
- The empirical evaluation demonstrates state-of-the-art performance on two established and challenging benchmarks for formal mathematics.
- The inclusion of a refutation-guided process for identifying unprovable goals is a practical feature that enhances the system's overall robustness.
### Weaknesses
- A potential weakness is that the experimental comparison in Table 1 lacks clarity, as the specific version of the Gemini model used for the main results is not explicitly stated, which could affect the fairness of the comparison against baselines using specified model versions.

---

> ### Author Rebuttal · Authors · 2026-03-30
>
> Thank you very much for your encouraging and thoughtful review. We sincerely appreciate your positive assessment of our contributions, especially the localized editing capability of EditableSketch and the overall effectiveness of the SketchRefine framework.
>
> ### Clarification on the Gemini version used in Table 1
>
> You are absolutely right that Table 1 reports **99.6%** while labeling the LLM simply as “Gemini”, which can be unclear. In our experiments, this “Gemini” result was obtained with a **mixed usage of two Gemini versions**:
>
> - **The vast majority of problems were solved using Gemini 2.5 Pro** (as stated in **Section 4.2**, 99.2% are solved by Gemini 2.5 Pro, outperforming Hilbert at 98.4% under the same settings).
> - We then additionally tried **Gemini 3.0 Pro Preview** on the remaining unsolved instances, which solved one extra problem (**`imo_2001_p6`**), bringing the overall pass rate to **99.6%** (the number reported in Table 1).
>
> So, the Table 1 “Gemini” figure corresponds to a **Gemini 2.5 Pro + Gemini 3.0 Pro Preview** mixture, with **Gemini 2.5 Pro accounting for most of the solved cases**.
>
> ### Revision for clarity
>
> We agree that Table 1 should explicitly reflect this. In the further revision version, we will revise the table caption to clearly state the mixed setup and to indicate that **most problems were solved by Gemini 2.5 Pro**, with **Gemini 3.0 Pro Preview** used only to cover the remaining hard cases.
>
> Thank you again for pointing this out—your feedback will help us make the experimental setting clearer and more reproducible.

---

> > ### Author Rebuttal · Reviewer_HA8x · 2026-03-31
> >
> > I have no further questions and will keep my positive score.

---

> > > ### Author Response · Authors · 2026-04-07
> > >
> > > Thank you for your thoughtful and constructive review. We appreciate your clear summary of our work and are grateful that you recognize the key contributions of our method, especially its support for localized edits. Your encouraging feedback is very helpful to us.

---

### Official Review · Reviewer_csa2 · 2026-03-09

**Soundness:** 2
**Presentation:** 3
**Significance:** 3
**Originality:** 3
**Overall Recommendation:** 4
**Confidence:** 4

**Summary:**

This paper solves the problem of editing fragility of Lean 4 scripts that a local modification can silently alter the surrounding proof context and cause downstream tactics to fail. To solve it, this paper proposes an editable proof-sketch structure named EditableSketch that supports in-place edits while preserving previously proved subgoals. It then introduces SketchRefine, a proof-generation framework that iteratively refines proof sketches through localized, incremental edits. The evaluation on the miniF2F-test and FormalMath-Lite datasets shows that SketchRefine outperforms the compared sketch generation methods.

**Compliance With Llm Reviewing Policy:**

Affirmed.

**Final Justification:**

The rebuttal has addressed most of my concerns.

**Key Questions For Authors:**

* What is the running time of SketchRefine for proving a theorem?
* Prompts for the LLMs/Provers in SketchRefine?

**Limitations:**

Yes.

**Strengths And Weaknesses:**

Strengths:
* This paper has insights into sketch generation for automated theorem proving by exploring the editing fragility.
* The proposed SketchRefine performs well on the miniF2F-test and FormalMath-Lite compared with previous sketch generation methods.
* The paper is well constructed and easy to follow.

Weaknesses:
* The evaluation is restricted to miniF2F and FormalMath-Lite, which are small-scale and have relatively easy theorems. Would like to see performance in additional datasets, e.g., PutnamBench, ProofNet.
* Reproducibility concern. It is unclear whether the authors would release their codes, as it is neither mentioned in the manuscript nor seen in the supplementary materials.
* Missing justification of the EditableSketch design. Would be better to add ablation studies of EditableSketch, for example, on step types/node types to help the understanding of the structure.
* Number of premises and their impact on the specialized prover in Section 4.3 seems merely an analysis of premise selection on a commercial LLM prover (DeepSeek Prover-V2 7B), which does nothing to support the effectiveness or analysis of the proposed method.

---

> ### Author Rebuttal · Authors · 2026-03-30
>
> We sincerely thank the reviewer for the careful reading and constructive feedback. Below we respond to each concern and question, and we will incorporate the corresponding revisions in the revised paper.
>
> ### W1: Limited evaluation datasets
>
> Thank you for the suggestion. We have added experiments on ProofNet and PutnamBench.
> **(1) ProofNet-test.** Results are:
>
> | Method                           | Pass Rate |
> | -------------------------------- | --------- |
> | ReProver                         | 13.8%     |
> | DeepSeek-Prover-V1.5-RL + RMaxTS | 25.3%     |
> | BFS-Prover-V2-32B                | 41.4%     |
> | **SketchRefine (ours)**          | **50.5%** |
>
> In this experiment, SketchRefine uses DeepSeek-V3.2-Exp as the general-purpose LLM and DeepSeek-Prover-V2-7B for proving subtheorems, with the reasoner tokens capped at 1M.
>
> **(2) PutnamBench (subset).** Due to resource constraints, we compare Hilbert and SketchRefine on a subset consisting of the first 50 problems in PutnamBench:
>
> | Method / Tokens         | 400K  | 800K  | 1.2M  | 1.6M  | 2.0M  |
> | ----------------------- | :---: | :---: | :---: | :---: | :---: |
> | Hilbert                 | 8.0%  | 14.0% | 18.0% | 22.0% | 24.0% |
> | **SketchRefine (ours)** | 20.0% | 32.0% | 32.0% | 36.0% | 38.0% |
>
> Both methods use DeepSeek-V3.2-Exp as the general-purpose LLM and DeepSeek-Prover-V2-7B for subtheorem proving. These results suggest SketchRefine generalizes beyond miniF2F and remains competitive on harder benchmarks.
>
> ### W2: Reproducibility and code release
>
> We agree this is important. Our codebase is currently being cleaned and documented. We will **release the code**, and it will include the **full prompts for all stages** to support reproducibility.
>
> ### W3: Missing justification / ablation for EditableSketch
>
> Thank you for the suggestion. We agree that the design choices of EditableSketch deserve more justification.
>
> Empirically, these node types are **commonly used** in practice. On miniF2F-test, among the **97** problems where we successfully constructed an EditableSketch, **42** include `CONSTRUCTION` nodes and **49** include `ASSUMPTION`/`CONCLUSION` nodes, indicating these components are not rare edge cases.
>
> We are now conducting node-type ablations to quantify their contributions, and will report the results in the revision. We also added an ablation on **SketchRefine’s Error Repair** step (see our reply to reviewer **oMKy, W2**) and will integrate the relevant findings into the revised paper.
>
> ## Weakness 4: Relevance of the premise-number analysis in Section 4.3
>
> Thank you for pointing this out. The goal of Section 4.3 is to motivate why we use `FROM` to **explicitly filter premises** for the specialized prover. In Fig. 5, we add random irrelevant premises to sub-theorems from the final EditableSketch and observe a **significant pass-rate drop even with only a few irrelevant premises**, despite the subproblem itself not becoming logically harder. This indicates that naively providing “all earlier conclusions” as premises can substantially hurt prover efficiency.
>
> We will clarify this motivation and its connection to the `FROM` design more explicitly in the revision.
>
> ### Q1: Running time of SketchRefine
>
> We measured end-to-end runtime on ProofNet-test with up to 64-way concurrency. Across 186 problems, the average time per problem is 3h 42m 51s. For the 95 successfully proved problems, the average time is 1h 31m 06s.
>
> We also observed that the general-purpose LLM (e.g. DeepSeek-V3.2-Exp) dominates the time cost. The network latency and API stability significantly affects wall-clock measurements. We will report these statistics clearly in the revised paper.
>
> ## Q2: Prompts used in SketchRefine
>
> We will release **all prompts** with the code. Here we provides a simplified prompt example of **Sketch Construction** stage:
>
> ```
> # Lean 4 Step-by-Step Proof Construction
>
> ## Task Overview
>
> You are a mathematical expert in Lean 4. Your task is to construct a step-by-step proof process for a given Lean 4 problem. The proof flow must follow the specific syntax format described below.
>
> ## Syntax Rules
> The Lean 4 problem will be provided as a set of declarations in the form `name : type`, along with the goal that needs to be proven. Your proof must use the following statement types:
>
> ### Statement Types
>
> #### 1. ASSUMPTION [name] : [type] REFER [names...]
>
> Represents the assumption that there exists an object named `name` of type `type`, or a proposition named `name` with content `type`.
>
> (...Detailed usage and Example)
>
> #### 2. CONSTRUCTION ([name1] : [type1]), ([name2] : [type2]), ... SATISFY [name] : [content] REFER [names...] FROM [i1] [i2] ...
> ...
>
> #### 3. DEDUCTION [name] : [content] REFER [names...] FROM [i1] [i2] ...
> ...
>
> #### 4. CONCLUSION [name] FROM ASSUMPTION [name_a] DEDUCTION [name_d]
> ...
>
> #### 5. GOAL [name]
> ...
>
> (...Examples and other Instructions)
> ```

---

> > ### Author Rebuttal · Reviewer_csa2 · 2026-04-03
> >
> > Thanks for the response.
> >
> > * Could you explain the budgets of each model on ProofNet and PutnamBench experiments?
> > * Looking forward to the node-type ablations.
> > * Could you further explain `FROM` design? I believe it is not introduced in your main paper.

---

> > > ### Author Response · Authors · 2026-04-06
> > >
> > > Thank you for your constructive acknowledgement and follow-up questions. We respond to each point below.
> > >
> > > ### Budgets of each model.
> > >
> > > We appreciate you noting that our earlier rebuttal did not clearly specify the budgets used by the compared baselines.
> > >
> > > - **ProofNet-test.** We report budgets exactly as stated in the original papers. ReProver [1] is reported with **Pass@1**. DeepSeek‑Prover‑V1.5‑RL + RMaxTS [2] uses a sampling budget of **$4\times 6400$**. BFS‑Prover‑V2‑32B [3] is described as **“accumulative”** in the original paper, without an explicit numeric budget. For SketchRefine, our budget is **1M  general-purpose LLM output tokens per problem** (for DeepSeek‑V3.2 this is approximately **$0.42**) as stated in our rebuttal.
> > >
> > > - **PutnamBench.** The “Tokens” row in our table denotes **general-purpose LLM output tokens**. As stated in the rebuttal, the budgets for **SketchRefine** and **Hilbert** are both **2M general-purpose LLM output tokens per problem**.
> > >
> > > Additionally, we would like to correct a minor issue in our previous rebuttal: both SketchRefine and Hilbert use the same  general-purpose LLM as in the main paper, i.e., DeepSeek‑V3.2.
> > >
> > > ### Node-type ablations.
> > >
> > > Following your suggestion, we conducted node-type ablations by modifying the prompting format and tested on MiniF2F-test (DeepSeek-V3.2 + DeepSeek-Prover-V2 7B): removing Assumption (and the paired Conclusion), removing Construction, and keeping only Deduction nodes. The pass rate vs.  general-purpose LLM output tokens is:
> > >
> > > | Method \  General-purpose LLM Output Tokens    | 200K  | 400K  | 600K  | 800K  | 1.0M  |
> > > | :--------------------------------- | :---: | :---: | :---: | :---: | :---: |
> > > | Only Deduction                     | 88.1% | 91.8% | 93.4% | 93.9% | 93.9% |
> > > | Deduction + Construction           | 88.1% | 91.8% | 92.2% | 93.4% | 93.9% |
> > > | Deduction + Assumption(Conclusion) | 86.9% | 90.6% | 93.0% | 93.0% | 94.3% |
> > > | All node types applied             | 90.2% | 93.4% | 93.9% | 94.7% | 95.5% |
> > >
> > > Removing any node type leads to a consistent drop in accuracy. Empirically, **Construction** helps avoid repeated expressions and enables concise handling of complex objects (e.g., introducing solutions of equations). **Assumption/Conclusion** reduces deeply nested "$\rightarrow$'' structures and is important for keeping the EditableSketch manageable.
> > >
> > > ### Clarification of the `FROM` design.
> > >
> > > We apologize for the confusing terminology. In our rebuttal, **`FROM` corresponds to "Premises" in Figure 2**, and **`REFER` corresponds to "Dependencies" in Figure 2**. They are syntax keywords defined in our prompting grammar. Section 3.1.1 provides further explanation, and the naming is described in Appendix B.1.
> > >
> > > Finally, in the revised version, we plan to move the current Figure 5 experiment to the appendix and replace it with the node-type ablation results above, as we agree this ablation is more informative. Thank you again for the helpful suggestion.
> > >
> > > [1] Yang, Kaiyu, et al. "Leandojo: Theorem proving with retrieval-augmented language models." *Advances in Neural Information Processing Systems* 36 (2023): 21573-21612.
> > >
> > > [2] Xin, Huajian, et al. "Deepseek-prover-v1. 5: Harnessing proof assistant feedback for reinforcement learning and monte-carlo tree search." (2024)
> > >
> > > [3] Xin, Ran, et al. "Scaling up multi-turn off-policy rl and multi-agent tree search for llm step-provers." (2025)

---

### Official Review · Reviewer_oMKy · 2026-03-10

**Soundness:** 3
**Presentation:** 3
**Significance:** 3
**Originality:** 3
**Overall Recommendation:** 3
**Confidence:** 4

**Summary:**

This paper investigates how to refine proof sketches in automated theorem proving with large language models (LLMs) to improve efficiency. The authors propose an editable proof-sketch structure that supports error correction while preserving already proven subgoals. In addition, they introduce a proof-generation framework that iteratively refines proofs. Experimental results demonstrate strong performance while consuming fewer tokens.

**Compliance With Llm Reviewing Policy:**

Affirmed.

**Final Justification:**

The authors did not fully address my points in their initial rebuttal. Without an appropriate motivating example, the work remains insufficiently justified. Without qualitative analysis, it is difficult to develop intuition about the real effects of the method beyond aggregate numerical results. Additionally, the lack of statistics on “sorry” placeholders makes the process less transparent.

The framework starts with sketches containing only “sorry” placeholders, and even if the final outputs contain none, intermediate steps should still include unresolved subgoals. If this is not the case, it would suggest a one-step process, which contradicts the claimed iterative nature of the approach. This issue is not adequately clarified.

For these reasons, I maintain my rating of 3.

**Key Questions For Authors:**

1. SketchRefine is designed as an iterative framework. How many iterations are typically required for each instance? It may be the case that many instances, especially simple theorems in miniF2F, do not require more than one iteration. In that case, the benefit of running the framework iteratively becomes questionable. A more detailed analysis of iteration counts would help better demonstrate the effectiveness of the proposed framework.

2. The authors define three edit types: Insertion, Update, and Deletion. How are these edit types distributed across different datasets within the proposed framework?

3. The results could benefit from more qualitative analysis of how the framework performs in practice. Could the authors provide a qualitative comparison between the proposed framework and other baselines, such as Hilbert? If the authors plan to include an example in the response phase, it would be helpful to select representative but relatively short examples.

**Limitations:**

The paper would benefit from a brief discussion of the limitations of the proposed framework. For example, the authors could discuss scenarios in which the approach may fail and provide a brief explanation of the underlying reasons.

**Strengths And Weaknesses:**

**Strengths**:

1. The proposed proof-sketch structure, EditableSketch, is novel and intuitively sound. By decomposing the proof into several steps, indexing these steps, and building connections between them, the proposed structure can facilitate iterative approaches in general while enhancing the reusability of previously proven subgoals across iterations.

2. The proposed iterative refinement framework, SketchRefine, which contains five steps, is well illustrated and clearly explained. The Theorem Retrieval technique designed to mitigate compatibility issues across different Lean versions is practically useful. The Refutation technique explicitly considers outliers when the provided theorem is unprovable, which adds further robustness to the proposed framework.

3. The proposed method is more efficient. Empirical evidence shows that the mechanism can reduce token consumption when using LLMs while maintaining strong performance in terms of pass rate.

**Weaknesses**:

1. The motivation regarding editing fragility, as illustrated in Figure 1, is relatively weak. The statement “have h : y = 0 := sorry” corresponds exactly to a subgoal that needs to be proved, and it is not difficult to prove such a subgoal using common tactics such as linarith. Therefore, the necessity of performing the editing in this example is not well justified.

2. The ablation study is relatively insufficient. Several steps are involved in the proposed framework, however, it is empirically unclear how much each step contributes to the overall improvement, such as the impact of the subgoal validation step. Removing certain steps would also reduce token consumption. It is possible that reallocating token budgets from some steps to others could achieve similar performance.

3. While I understand that pass rate is a common metric for evaluating automated theorem proving systems, it can potentially be manipulated through the use of the sorry placeholder. When a formal proof contains many subgoals, some of them may remain unproven. The proof-sketch introduced in this paper uses sorry extensively, yet this issue is not evaluated in the evaluation.

---

> ### Author Rebuttal · Authors · 2026-03-30
>
> Thank you for the careful reading and constructive feedback. We appreciate the positive assessment of EditableSketch that it is novel and intuitively sound, and we respond to your concerns point-by-point below.
> ### W1. Motivation of editing fragility (Figure 1)
>
> We agree that the original Figure 1 example was too simple and did not convincingly demonstrate *editing fragility*. In the revision, we will replace it with a stronger Lean4 example where a seemingly local correction triggers unexpected downstream failures due to non-local proof-state coupling:
>
> ```lean
> theorem ex_thm (x y : ℕ) (h₁ : x + (x - 2) = 0) (h₂ : y = x) :
>   y = 0 ∨ y = 1 := by
>   have h₃ : x = 1 := by sorry    -- Impossible to prove
>   rw [h₃] at h₂
>   apply Or.inr
>   exact h₂
> ```
>
> Under Lean4’s truncated subtraction on `ℕ`, `h₁ : x + (x - 2) = 0` forces `x = 0`, so the subgoal `x = 1` is actually unprovable. One may try to “simply” edit `h₃` to `x = 0`:
>
> ```lean
> theorem ex_thm (x y : ℕ) (h₁ : x + (x - 2) = 0) (h₂ : y = x) :
>   y = 0 ∨ y = 1 := by
>   have h₃ : x = 0 := by
>     rw [add_eq_zero] at h₁
>     exact h₁.left
>   rw [h₃] at h₂
>   apply Or.inr
>   exact h₂  -- tactics fail due to changed downstream state
> ```
>
> Even though the edit appears local, it changes the proof state in a way that breaks a later line.
>
> ### W2. Insufficient ablation study
>
> Thank you for suggesting a clearer attribution of gains to individual steps. In the revision, we will add an ablation targeting **Subgoal Validation** (and the dependent **Error Repair** stage). On miniF2F-test (DeepSeek-Prover-7B + DeepSeek-V3.2-Exp), we observe consistent drops in pass rate when the validation/repair mechanism is removed:
>
> | Method                                     | 200K  | 400K  | 600K  | 800K  | 1.0M  |
> | :----------------------------------------- | :---: | :---: | :---: | :---: | :---: |
> | SketchRefine w/o Error Repair & Validation | 86.5% | 88.5% | 91.0% | 91.8% | 92.2% |
> | SketchRefine w/ Error Repair & Validation  | 90.2% | 93.4% | 93.9% | 94.7% | 95.5% |
>
> These results indicate that validation/repair contributes materially across token budgets.
>
> ### W3. Pass rate and the `sorry` placeholder
>
> We appreciate this important concern. In our evaluation, `sorry` is **not** permitted in the final proofs used to compute pass rate: we run Lean under a strict checking setting that rejects any code containing `sorry`.
>
> Importantly, **EditableSketch uses `sorry` only as a placeholder inside the decomposed sub-theorems that are intended to be discharged by the specialized prover**. Once the prover completes these sub-theorem proofs, the corresponding `sorry` placeholders are naturally eliminated. As a result, the final end-to-end proof artifact contains **no `sorry`**. We will clarify this explicitly in the revised paper.
> ### Q1. How many iterations are typically required?
>
> Thank you—this is a valuable request. We analyzed DeepSeek-V3.2-Exp + DeepSeek-Prover-V2 7B on miniF2F-test. Iteration is non-trivial: **41** problems required at least one EditableSketch edit; across these, there were **177** Error Repair calls (4.3 per edited problem on average) and **226** Hard Subgoal Decomposition calls (5.5 per edited problem on average). We will add a more detailed distribution in the revision.
>
> ### Q2. Distribution of edit types (Insertion/Update/Deletion)
>
> Thank you for asking. In the same miniF2F-test run, we observed **403** total EditableSketch edits, comprising **2,776 insertions**, **664 updates**, and **1,116 deletions** (insertions dominate, with deletions/updates also frequent).
>
> ### Q3. Qualitative comparison with baselines
>
> We agree that qualitative analysis would strengthen the paper. In the revision, we will include a short representative case study from `aime_1999_p11`. In this problem, Hilbert introduces (expA is a complex expression):
> ```lean
> theorem h_sum_as_cot_aime_1999_p11 : expA = Real.cos(Real.pi/72)/Real.sin(Real.pi/72)
> theorem h_tan_eq_aime_1999_p11 (m:ℚ) (h₁: expA = Real.tan(m*Real.pi/180)) :
>   Real.tan(m*Real.pi/180) = Real.tan(35*Real.pi/72)
> ```
> Since `h_tan_eq_aime_1999_p11` does not reuse `h_sum_as_cot_aime_1999_p11` as an assumption, Hilbert further creates an additional subgoal:
> ```lean
> theorem h_sum_eq_h_tan_eq_aime_1999_p11 : expA = Real.tan (35 * Real.pi / 72)
> ```
> which is essentially the same difficult subtheorem, leading to repeated effort and substantial extra token cost. In contrast, EditableSketch detects the missing dependency and repairs the sketch without repeating proof.
>
> ### Limitations
>
> We will add a brief limitation paragraph in revision. In particular, EditableSketch primarily helps a general LLM **localize errors and repair proof structure** using Lean feedback. It does **not** fundamentally increase the LLM’s ability to discover proofs for problems beyond its capacity. For example, if the LLM cannot find a viable solution strategy (e.g., hard miniF2F instances like `imosl_2007_algebra_p6`), editable refinement alone may not resolve the failure.

---

> > ### Author Rebuttal · Reviewer_oMKy · 2026-03-31
> >
> > Thank the authors for their rebuttal. I appreciate the effort. However, my concerns have not been sufficiently addressed:
> >
> > **On the examples.**
> >
> > - The newly introduced motivational example is mathematically inconsistent. Translating the formal statement back into standard mathematics yields a statement over natural numbers where simple reasoning implies $x=1$, whereas the Lean4 formulation effectively leads to $x=0$. While I understand this is caused by Lean4's design of operations on $\mathbb{N}$, this discrepancy arises from the choice of $\mathbb{N}$ instead of $\mathbb{Z}$. This example in fact highlights a mismatch between informal mathematics and its formalization. Rather than clarifying the approach, this example introduces additional confusion. A better choice would have been either a correctly aligned formulation or a representative example from the benchmark where both logics coincide.
> >
> > - Furthermore, the qualitative example does not demonstrate the contribution of EditableSketch. It does not show the generated proof, nor how EditableSketch improves over Hilbert baseline. As a result, it provides little insight into the claimed advantages of the approach.
> >
> > **On "sorry" placeholders.**
> >
> > - The rebuttal still does not adequately address how "sorry" placeholders are handled. The details of rejecting codes containing "sorry" are not provided. While the final proof may not contain "sorry", the iterative nature of the framework implies that intermediate steps can include them. It is therefore important to track how "sorry" placeholders are eliminated over time to ensure genuine progress.
> >
> > - In the proof of each sub-theorem, "sorry" placeholders may still appear. This issue is not merely a matter of theorem complexity. It reflects the fundamental limitation that the prover may fail to complete certain sub-theorem proofs. The current argument assumes that all sub-theorems will eventually be successfully proven, which is a strong assumption and not justified given the known limitations of LLM-based provers. If some sub-theorems remain unresolved, the corresponding "sorry" placeholders may persist, directly affecting the correctness and completeness of the final proof. As a result, this weakens the overall argument and raises concerns about the generalizability of the proposed framework, particularly when applied to more complex theorems where incomplete sub-proofs are more likely.
> >
> > Overall, the rebuttal does not fully resolve my concerns. While I understand the time constraints associated with the rebuttal process, the issues raised could likely have been addressed with relatively modest additional effort. Given that these points remain insufficiently clarified, I would prefer to maintain my original assessment of the paper.

---

> > > ### Author Response · Authors · 2026-04-01
> > >
> > > Thank you again for your detailed follow-up. We sincerely agree that our previous rebuttal did not sufficiently address your concerns. Below we respond more concretely.
> > >
> > > ## On the examples
> > >
> > > Following your suggestion, we selected a classic miniF2F benchmark problem `imo_2006_p6`. The standard solution uses the homogeneity of the inequality to assume that $(a^2 + b^2 + c^2 = 1)$. A straightforward Lean formalization skeleton looks like:
> > >
> > > ```lean
> > > def f (x y z : ℝ) : ℝ :=
> > >   x*y*(x^2-y^2)+
> > >   y*z*(y^2-z^2)+
> > >   z*x*(z^2-x^2)
> > >
> > > theorem imo_2006_p6 (a b c : ℝ) : f a b c ≤
> > > 9 * Real.sqrt 2 / 32 * (a ^ 2 + b ^ 2 + c ^ 2) ^ 2 := by
> > >   let t := Real.sqrt (a ^ 2 + b ^ 2 + c ^ 2)
> > >   let na := a / t
> > >   let nb := b / t
> > >   let nc := c / t
> > >   have h0 : t ^ 4 ≥ 0 := by ...
> > >   have h1 : na ^ 2 + nb ^ 2 + nc ^ 2 = 1 := by sorry		-- impossible
> > >   have h2 : (a ^ 2 + b ^ 2 + c ^ 2) ^ 2 = t ^ 4 := by ...
> > >   have h3 : f na nb nc ≤ 9 * Real.sqrt 2 / 32 := by ...
> > >   have h4 : f a b c = f na nb nc * t ^ 4 := by ...
> > >   rw [h4, h2]
> > >   apply mul_le_mul_of_nonneg_right
> > >   exact h3
> > >   exact h0
> > > ```
> > >
> > > **Why this fails and why editing is nontrivial in raw Lean.**
> > >
> > > If `a = b = c = 0`, then `t = 0` and the normalization is invalid. This appears in the code as the lemma **`h1`** being fundamentally **unprovable** unless we add the missing premise `t ≠ 0` . The correct formal proof must do a case split on whether `t` is zero. However, modifying the raw Lean script to introduce such branching often requires complex intrusive edits (`by_cases`, `cases`), and because the context changes globally, previously proven sub-lemmas (`h0`, `h2`–`h4`) may need to be re-proved.
> > >
> > > **How EditableSketch makes this edit localized and structurally natural.**
> > > EditableSketch can express the required change by inserting two branches (for `t ≠ 0` and `t = 0`) at the sketch level. Let `goal_exp` abbreviate the target goal
> > > `f a b c ≤ 9 * Real.sqrt 2 / 32 * (a ^ 2 + b ^ 2 + c ^ 2) ^ 2`.
> > > Then the edit can be represented as:
> > >
> > > ```
> > > CONSTRUCTION (t : ℝ) SATISFY (h_t : t = Real.sqrt (a ^ 2 + b ^ 2 + c ^ 2)) REFER a b c FROM
> > > [ASSUMPTION ht_neq0 : t ≠ 0 REFER t FROM]
> > > CONSTRUCTION (na : ℝ) SATISFY (h_na : na = a / t) REFER a t FROM
> > > ...
> > > DEDUCTION h1 : na ^ 2 + nb ^ 2 + nc ^ 2 = 1 REFER na nb nc FROM h_na h_nb h_nc h_t [h_tneq0]
> > > ...
> > > [DEDUCTION h5 : goal_exp REFER ... FROM ...]
> > > [CONCLUSION h6 FROM ASSUMPTION ht_neq0 DEDUCTION h5]
> > > [ASSUMPTION ht_eq0 : t = 0 REFER t FROM]
> > > [DEDUCTION h7 : goal_exp REFER ... FROM ...]
> > > [CONCLUSION h8 FROM ASSUMPTION ht_eq0 DEDUCTION h7]
> > > [DEDUCTION hgoal : goal_exp REFER a b c FROM h6 h8]
> > > ```
> > >
> > > Here, the bracketed parts are **newly added** compared to the pre-edit sketch.
> > >
> > > **What this demonstrates.**
> > >
> > > 1. The modification matches mathematical intuition.
> > > 2. The heavy Lean-level refactoring is avoided: the case split is expressed declaratively, while specialized small models handle the corresponding Lean syntax.
> > > 3. Edit locality: aside from `h1` (which must be reproved), other already-completed subgoals can remain unchanged.
> > >
> > > We agree with your critique that our prior qualitative example did not show enough. In the revision we will include this example to make the editing advantage explicit.
> > >
> > > ## On `sorry` placeholders
> > >
> > > **Detecting and rejecting code that contains `sorry`**
> > >
> > > For the prover used to discharge sub-theorems, a candidate proof that contains `sorry` does **not** pass verification.
> > >
> > > Concretely, we read all messages returned by the `kimina-server` API from Lean compilation:
> > >
> > > - If `severities` contains an `error`, the proof is rejected as syntactically invalid.
> > > - If it contains a `warning` indicating that a statement uses `sorry`, the proof is also rejected.
> > >
> > > This implementation is identical to Hilbert’s handling (the `read_client_response` in `https://github.com/apple/ml-hilbert/blob/main/src/tools/proof_utils.py`).
> > >
> > > **Tracking `sorry` across iteration**
> > >
> > > You are correct that iterative frameworks must be explicit about intermediate artifacts. In our framework, `sorry` is used only as a **temporary placeholder** in the scaffolding code that materializes sub-theorem statements.
> > >
> > > A single EditableSketch roughly corresponds to code of the following form:
> > >
> > > ```lean
> > > theorem h1 ... := sorry
> > > theorem h2 ... := sorry
> > > theorem ori_problem ... :=
> > >   have h11 := h1
> > >   have h12 := h2 h11
> > >   exact h12
> > > ```
> > >
> > > Key points:
> > >
> > > 1. The proof of `ori_problem` is constructed by fixed composition rules and does **not** introduce any `sorry`.
> > > 2. Sub-theorem statements contain `sorry` will be accepted.
> > > 3. SketchRefine then calls the prover to replace each `sorry` body with a fully verified Lean proof one by one. As explained above, any attempt that still triggers a `sorry` warning is rejected.
> > > 4. If some sub-theorems cannot be proven, the framework will edit the sketch. We do **not** claim success unless all sub-theorems are discharged with no `sorry` warnings and no errors.
> > > 5. Therefore, for any final accepted output, `ori_problem`, `h1`, `h2`, etc. all have proofs that contain **no** `sorry`.

---

### Official Review · Reviewer_6KVh · 2026-03-11

**Soundness:** 2
**Presentation:** 3
**Significance:** 2
**Originality:** 2
**Overall Recommendation:** 4
**Confidence:** 4

**Summary:**

The authors propose EditableSketch, an editable proof-sketch structure supporting localized in-place edits (insertion, update, deletion) with explicit dependency tracking between proof steps, and build the SketchRefine iterative proof-generation framework on top of it.
Experiments on MiniF2F-test and FormalMath-Lite show state-of-the-art pass rates (99.6% and 76.0% respectively) and improved token efficiency compared to baselines like Hilbert and DeepSeek-Prover-V2-671B. The core contributions include the EditableSketch structure, the SketchRefine framework, and empirical validation on standard ATP benchmarks.

**Compliance With Llm Reviewing Policy:**

Affirmed.

**Final Justification:**

They have shown the connection between their approach to RL. It resolves my concern.

**Key Questions For Authors:**

Can you provide systematic experimental results of EditableSketch/SketchRefine on large-scale benchmarks and train it on large models?
I also want to see it in training with RL. If you can show training using this approach on RL with an 8B, it would be interesting.

**Limitations:**

yes

**Strengths And Weaknesses:**

# Strenghts

Solid Empirical Validation: The authors conduct comprehensive experiments on two benchmarks (MiniF2F-test, FormalMath-Lite). Results are clearly presented and demonstrate consistent improvements in pass rate and token efficiency over existing methods.

# Weaknesses

- There are many such papers showing seemingly novel search approaches. However, none of those show the potential that their approaches can scale with larger models and RL. It is highly unclear whether this approach is helpful for training a math LLM using RL. However, most frontier labs choose to perform RL to train SOTA math LLMs. Hence, I highly doubt the value of this paper.
- The approach is not novel. As I explained, many papers have tried to decompose Lean subgoals or design new proof search procedures.

---

> ### Author Rebuttal · Authors · 2026-03-30
>
> Thank you for the thoughtful review and for raising important questions about scalability, RL compatibility, and novelty. We agree that demonstrating usefulness in the large model and RL regime is increasingly relevant for frontier-level math LLMs, and we appreciate the opportunity to clarify where our work fits.
>
> ## Scaling to RL
>
> We agree that many recent gains in mathematical reasoning come from RL, and we did not include RL training results in this submission. That said, our proposed EditableSketch is structurally well-aligned with an RL formulation:
>
> - The pair (problem, current EditableSketch) naturally defines an RL **state**.
> - **Edits** to the sketch, including Insertion, Update, and Deletion of nodes, naturally define **actions**.
> - A policy can be trained to propose the next sketch modification, iterating until each remaining subproblem can be discharged by a simpler prover.
>
> This differs from most prior training setups, which predict the next **Lean tactic** given a Lean goal state (e.g., BFS-style tactic provers), or iteratively rewrite full **Lean code**. Both are often tightly coupled to Lean’s surface-level tactic choices. In contrast, EditableSketch optimizes over a global proof sketch with explicit dependencies, encouraging the model to operate at the level of mathematical structure and decomposition rather than low-level Lean code details. We believe this makes it a promising interface for RL: the action space is explicitly defined and localized.
>
> Concretely, our next step is to explore **training models to edit EditableSketch** (including with RL). We appreciate the suggestion of an RL experiment on an 8B model; we agree it would be an informative direction and plan to pursue it.
>
> ## Large-scale benchmark results
>
> In this work we focused on standard ATP-style formal benchmarks to provide controlled comparisons and clear ablations. In addition, as included in our response to reviewer **csa2**, we have added new experimental results on ProofNet-test and on a subset of PutnamBench to further assess scalability beyond the two main benchmarks. We acknowledge that fully large-scale evaluation and RL training on larger models remains an important next step, and we plan to expand both benchmark coverage and training regimes in future work.
>
> ## Novelty relative to prior decomposition work
>
> We agree that “decompose subgoals proof search procedures” is an active area, and we do not claim to be the first to use decomposition in Lean. Our intended novelty is **not decomposition alone**, but the **EditableSketch abstraction** and its associated **localized in-place edit operations** with explicit dependency tracking. While prior approaches often operate on Lean goal states with tactic-level steps, or whole-program rewrites, our method explicitly maintains and edits a dependency-aware proof sketch as a first-class object. This representation enables fine-grained, localized modifications without regenerating entire proofs and is designed specifically to support iterative refinement workflows.
>
> Finally, we want to emphasize that we share the reviewer’s concern: if an approach cannot plausibly extend to the large-model and RL regime, its long-term value is limited. Our main point is that **EditableSketch was designed in a way that is naturally compatible with RL**, even though we have not yet executed the full RL training pipeline in this submission. We will make this positioning clearer in the paper and outline a concrete RL training protocol as future work.

---

> > ### Author Rebuttal · Reviewer_6KVh · 2026-04-03
> >
> > I’d like to see a mathematical formulation of their approach, expressed in a reinforcement learning (RL) framework, included in the appendix.

---

> > > ### Author Response · Authors · 2026-04-06
> > >
> > > Thank you for the suggestion to include a more rigorous mathematical formulation of EditableSketch in an RL framework. We agree this would clarify how the method can be integrated with RL-based training, and we will add the following formulation to the appendix.
> > >
> > > Given a Lean theorem $T$, we define a theorem-specific MDP
> > > $$
> > > \mathcal{M}_T=\langle \mathcal{S},\mathcal{A},P,r,\gamma\rangle.
> > > $$
> > >
> > > ### State
> > >
> > > A state $s_t\in\mathcal{S}$ is an EditableSketch represented as a directed graph:
> > > $$
> > > s_t=(V_t,E_t,x_t,\phi_t,n_t,p_t).
> > > $$
> > > Here $V_t$ is the set of sketch nodes. For each node $v\in V_t$: $n_t(v)$ is the number of terms constructed at $v$; $\phi_t(v,i)$ denotes the Lean type of the $i$-th constructed term at $v$; and $p_t(v)\in \{0,1\}$ indicates whether node $v$ is currently proved by the prover. The edge set $E_t\subseteq V_t\times V_t$ encodes dependencies; for each edge $e\in E_t$, $x_t(e)$ is a discrete edge label (e.g., premise, dependency).
> > >
> > > ### Action
> > >
> > > The action space $\mathcal{A}$ is the union of three edit operators:
> > >
> > > - **Add a node**
> > >   $$
> > >   a_t=\mathrm{Add}(v,\mathrm{edges},x,\phi,n),
> > >   $$
> > >
> > >
> > > - **Delete a node**
> > >   $$
> > >   a_t=\mathrm{Delete}(v),
> > >   $$
> > >
> > >
> > > - **Update a node**
> > >   $$
> > >   a_t=\mathrm{Update}(v,\mathrm{edges},x,\phi,n),
> > >   $$
> > >   where the parameters specify the node, edges, labels, and metadata to be inserted or revised.
> > >
> > > ### Transition
> > >
> > > State transitions follow
> > > $$
> > > s_{t+1}\sim P(\cdot\mid s_t,a_t).
> > > $$
> > >
> > > Operationally, $P$ is induced by applying the edit $a_t$ to the current sketch, after which we invoke the prover and Lean verifier to compute the proof-status indicators $p_{t+1}$.
> > >
> > > ### Reward
> > >
> > > We could use a sparse terminal reward consistent with common formal-proof RL setups:
> > > $$
> > > r(s_t,a_t,s_{t+1})=
> > > 1 \quad \text{if}\ \forall v\in V_{t+1},\, p_{t+1}(v)=1,\  \text{else} \  0
> > > $$
> > >
> > > ### Remarks
> > >
> > > This shows that EditableSketch naturally induces an RL environment: a base LLM can propose an initial coarse sketch, and an RL-trained policy can iteratively refine it via edits to reach a verified proof. We also acknowledge that the action, especially edits involving $\phi$ and term construction, may require further design (e.g., AST-level generation or constrained decoding with a formal grammar). RL designs are not the focus of this submission, but we appreciate this valuable direction and will include the above RL formulation in the appendix.

---

### Decision · Program_Chairs · 2026-04-30

**Decision:**

Accept (regular)

**Comment:**

This paper investigates how to refine proof sketches in automated theorem proving with large language models to improve efficiency. It proposes an editable proof-sketch structure, EditableSketch, that supports in-place edits while preserving previously proved subgoals. It then introduces SketchRefine, a proof-generation framework that iteratively refines proof sketches through localized, incremental edits. The evaluation on the miniF2F-test and FormalMath-Lite datasets shows that SketchRefine outperforms the compared sketch generation methods. The reviewers appreciate the importance of the problem, the solution and the empirical results, they also raised some concerns regarding novelty, ablations, qualitative differences with other approaches, and the confusing regarding the "sorry" placeholder which warrants a more detailed discussion.